# Novel Formula of Antiprotozoal Mixtures

**DOI:** 10.3390/antibiotics11070913

**Published:** 2022-07-07

**Authors:** Hubert Iwiński, Jacek Łyczko, Henryk Różański, Antoni Szumny

**Affiliations:** 1Department of Food Chemistry and Biocatalysis, Wrocław University of Environmental and Life Sciences, ul. C.K. Norwida 25, 50-375 Wrocław, Poland; jacek.lyczko@upwr.edu.pl (J.Ł.); antoni.szumny@upwr.edu.pl (A.S.); 2AdiFeed Sp. z o.o., Opaczewska, 02-201 Warsaw, Poland; rozanski@rozanski.ch; 3Laboratory of Industrial and Experimental Biology, Institute for Health and Economics, Carpathian State College in Krosno, Rynek, 38-400 Krosno, Poland

**Keywords:** antiprotozoal, antiparasitic activity, essential oils, LD_50_, phytochemistry, bioactivity, terpenes, phytoncides

## Abstract

Antimicrobial resistance (AMR) is becoming more common in both bacteria and pathogenic protozoa. Therefore, new solutions are being sought as alternatives to currently used agents. There are many new ideas and solutions, especially compounds of natural origin, including essential oils. In the present study, the antiprotozoal activity of a mixture of essential oils (eucalyptus, lavender, cedar and tea tree), organic acids (acetic acid, propionic acid and lactic acid) and metal ions (Cu, Zn, Mn) were tested. As a model, protozoans were selected: *Euglena gracilis*, *Gregarina blattarum*, *Amoeba proteus*, *Paramecium caudatum*, *Pentatrichomonas hominis*. The tested concentrations of mixtures were in the range of 0.001–1.5%. The analyses show unexpected, very strong protozoicidal activity of combinations, presenting the synergy of compounds via determination of LD_50_ and LD_100_ values. Obtained mixtures showed significantly higher activity against protozoans, compared to chloramphenicol and metronidazole. Most of the analyzed samples show high antiprotozoal activity at very low concentration, in the range of 0.001–0.009%. The most effective combinations for all analyzed protozoans were the cedar essential oil and tea tree essential oil with a mixture of acids and manganese or zinc ions. Innovative combinations of essential oils, organic acids and metal ions are characterized by very high antiprotozoal activity at low doses, which, after further investigation, can be applicable for control of protozoan pathogens.

## 1. Introduction

Each year, more than 15 million people die worldwide due to infectious diseases caused by various pathogens including bacteria, protozoa, viruses or fungi [1]. One of the main causes of death, besides bacterial and viral diseases, are diseases caused by protozoa such as *Plasmodium spp., Cryptosporidium spp., Leishmania spp.* and *Trypanosoma spp*. [2].

The most deadly diseases caused by protozoa undoubtedly include malaria, human African trypanosomiasis (HAT, sleeping sickness), Chagas disease, Visceral leishmaniasis (VL, kala azar), toxoplasmosis, Naegleriasis (PAM—primary amoebic meningoencephalitis), babesiosis or cryptosporidiosis [2,3,4]. The most common signs of protozoan infection include diarrhea, fever, malaise or hepatosplenomegaly [2,5].

Currently, there are three main methods of controlling protozoan parasites: prevention and containment of protozoan vectors, vaccination and antibiotic-based pharmacotherapy [2,6]. Based on the example of malaria, we know that the primary and so far most effective weapon in the fight against parasitic protozoa is prevention and elimination of vectors, which are most often mosquitoes, ticks or bedbugs. To reduce the risk of infection, many international and national guidelines recommend the use of insecticide-treated bed nets (ITNs), long-lasting insecticidal nets (LLINs), indoor residual spraying (IRS) or both methods together. Moreover, control of vector abundance also contributes significantly to reduce the spread of pathogens. The most commonly used method of controlling protozoa is the use of pharmaceuticals. Most often, these are single active substances that show antimicrobial activity. These include nifurtimox, pentamidine, quinine sulfate, fexinidazoles, benznidazoles, artemisinin derivatives and chloroquine, among others. However, most of them have been marketed for completely different purposes and medical indications. Mixed therapies combining two or more pharmacological agents, for example, nifurtimox-eflornithine (NECT), artemether-lumefantrine or quinine sulfate with doxycycline, tetracycline or clindamycin are also commonly used to optimize effects [5,7,8,9,10,11].

Mixed therapies are also a response to increasing resistance of microorganisms to available antibiotics. This is currently a significant problem that affects people all over the world. Its genesis is very complex and is influenced, among others, by high mobility and the possibility of movement all over the world, inadequate hygienic and sanitary conditions or the excessive use of antibiotics both in humans and animals [12,13]. As mentioned above, reducing pathogenic protozoa, especially those showing resistance, is through combination therapy. However, without clinical trials using similar combinations, it is very difficult to estimate their cytotoxicity and interaction in the body.

The lack of new and effective antibiotics causes the need for another alternative. This leads very often to developing an effective method to fight protozoa. The object of interest of researchers around the world are currently natural compounds extracted or obtained from plants [14,15,16]. Compounds showing antimicrobial activity include phenolic compounds, terpenes, sulfur glycosides or alkaloids [17].

Essential oils are characterized by a very high antimicrobial and antiprotozoal potential [18,19,20]. They are volatile mixtures, most of which are scented, of various substances of mainly plant origin. They are obtained via steam distillation from various fragments of plant leaves, flowers, fruits, buds, bark, seeds and even roots. They are usually stored in glandulars [21]. The growing interest in natural solutions, being an alternative to the currently used, generates new studies and reports in the literature. Papers or patents present that EOs are characterized by great antimicrobial potential. In vitro studies performed show very strong antiprotozoal activity of, e.g., tea tree oil (*Melaleuca alternifolia* (Maiden & Betche) Cheel) [22], lavender (*Lavandula angustifolia* Mill.) [23], thyme (*Thymus vulgaris* L.) [24], catnip (*Nepeta cataria* L.) [25], yarrow (*Achillea millefolium* L.), clove (*Syzygium aromaticum* L.), basil (*Ocimum basilicum* L.) [18], *Lippia* sp. [26], peppermint (*Mentha piperita* L.) [27] and rosemary (*Rosmarinus officinalis* L.) [28]. The current interest in natural alternatives to the available therapies is generating an increasing amount of scientific research. A growing number of studies are also investigating the antiprotozoal properties of individual components of essential oils, such as monoterpenoids and sesquiterpenoids as well as their alcohol, ester or ketone derivatives [29].

Organic acids are commonly known compounds used in food, cosmetic, feed and pharmaceutical markets as acidifiers, stabilizers, acidity regulators or preservatives. The last feature has been used widely through the ages for food prevention. That property is mainly based on the lowering of the pH, which results in the inhibition of microbials. Organic acids show very good results in broilers and fish production, by their antibacterial and antiprotozoal properties [30,31,32,33].

The combination of metals, e.g., iron, cobalt, nickel, gallium, copper, gold or silver with drugs are well known and very effective. Complex of Ru(II) chloroquine was one of the first tested antiprotozoal drugs combined with metal ions and showed a much better result than chloroquine itself. Other examples tested and evaluated for their antiprotozoal properties can be auranofin, triethylphosphine gold(I) chloride, cisplatin, 2-mercaptopyridine N-oxide complexes with Pt(II), Au(I) and Pd(II), or one of the most efficient for organometallic compounds, ferroquine [34,35,36]. Moreover, not only do the combining synthetic drugs and metals have great potential. Natural compounds such as essential oils, alkaloids and phenols in combination with metal ions are also investigated by researchers [37].

The mode of action of metal ions, mostly, is to impair the proper function of the cell membrane. They may be incorporated with the cell membrane, modulate ion channels, disrupt proton transfer or electrostatically interact with charges on the membrane surface. Metal ions can also affect various cell processes within cytoplasm such as inhibition of enzymes and proteins, catalyze the n of oxygen and hydroperoxide radicals and interrupt nutrient uptake [37,38,39].

The aim of this study was to investigate antiprotozoal properties of an innovative mixture of essential oils (eucalyptus, lavender, cedar and tea tree), organic acids (acetic, propionic, lactic) and metal ions (Cu, Zn and Mn). Previous research and scientific papers showed antiprotozoal properties of all components, but there were no scientific data about the proposed combinations.

## 2. Results

### 2.1. Antiprotozoal Activity

In the study, antiprotozoal properties were analyzed for single components of the mixtures: 48 combinations (4 essential oils with 3 metals and 3 organic acids and their mixtures). All obtained combinations are presented in Table 1. Chloramphenicol and metronidazole were used as standard substances. The obtained results are presented in Table 2, Table 3, Table 4, Table 5, Table 6, Table 7, Table 8 and Table 9. They are characterized by variable efficiency depending on the combination used and the protozoan species. The combinations containing an essential oil, a single acid and one of the selected metals showed much stronger antiprotozoal properties than the single components, and in some cases, than the antibiotics used for comparison. Among these combinations, the combination of tea tree essential oil, with propionic acid and manganese (TPMn) showed the highest potential. TPMn showed LD_50_, at a very low amount—0.01%. That result is much better than the reference antibiotics—chloramphenicol and metronidazole. It was also observed that the strongest activity independently of the used oil was observed for propionic acid and manganese ions.

The lowest antiprotozoal values against analyzed protozoa were obtained for innovative combinations of essential oil with the mixture of organic acids and selected metal. They showed an activity almost ten times stronger than that of analogous mixtures containing a single organic acid and an activity almost a hundred times stronger than that of single essential oils. Many of the proposed combinations showed activity against protozoa in the lowest tested concentration—0.001%. Most of the mixtures were very effective against *Euglena gracilis*. This protozoa was the most vulnerable form of all tested organisms. The concentration 0.001% was equal LD_50_ for 4 combinations, tea tree essential oil, mixture of acids and copper (TMCu); tea tree essential oil, mixture of acids and manganese (TMMn); cedar essential oil, mixture of acids and copper (CMCu); lavender essential oil, mixture of acids and manganese (LMMn). The highest concentration for the essential oil, mixture of acids and metal ions was obtained for the tea tree essential oil, mixture of acids and manganese (TMMn)—0.009%. Nevertheless, all the combinations with manganese ions showed the best results for all protozoans. However, it should be noted that the worst result for the combination of essential oil, mixture of acids and metal, is more than 10 times better than for the reference substance, which in this case is metronidazole. The values obtained for the remaining combinations were characterized by significantly lower effective doses than the reference substances, chloramphenicol or metronidazole. The mixtures with highest potential and lowest LD_50_ and LD_100_ values can be found in Table 10. Comparison between the most effective compositions, analysis of variance (ANOVA) using Tukey’s test, are presented in the Appendix A.

### 2.2. GC-MS Analysis of the Compositions

The chromatograms (presented in Appendix A Appendix A) show the chemical composition of the four essential oils used. Presented in Appendix A proves the typical mixtures for these essential oils’ composition. Predominated compounds for lavender EOs are linalyl acetate, linalool, 1-terpinen-4-ol and eucalyptol. In case of eucalyptus oil, 80.7% of the sum of the investigated compounds is an 1,8-cineol (eucalyptol) together with p-cymene 9.8%. Tea tree oil was rich in terpineols (mainly 1-terpinen-4-ol ~40%) with corresponding terpinenes (respectively γ–15.4% and α–9%). Finally, cedar oil was himachalene chemotype (β-isomer, 27.3% and α, with 9.1% content) together with sesquiterpenoic atlantone (isomer Z, 12.3% and trans-α 9.6%).

Detailed composition of investigated EOs are presented in Appendix A.

## 3. Discussion

The results of the in vitro studies presented below confirm the antiprotozoal properties of blends which contained in their composition essential oils (eucalyptus, cedar, lavender or tea tree) with organic acids (acetic, propionic, lactic) and metal ions (Cu, Zn, Mn).

The combination containing essential oils, organic acids and metal ions was a concept that occurred after several years of research on natural alternatives for antibiotics. Moreover, and what is very important, all of the used components are allowed to be used in food and feed. The idea corresponds strongly with the scientific results obtained by other researchers. Components used in this study were chosen by their antiprotozoal activity and different polarity [20,40,41,42].

To the best of our knowledge, presented in this paper, combinations of EOs, acids and metal ions were tested for the first time. The object of investigations that have been conducted, are usually combinations of two or more essential oils [43], metals [38,42], antibiotics [44,45], organic acids [46], active compounds occurring in essential oil [22], essential oils with other compounds of natural origin [47] or other compounds of plant origin e.g., alkaloids [48,49,50], triterpene saponins [51] or chalcones and flavonoids [41]. The proposed combination is characterized by innovation and, most importantly, high efficiency.

Eucalyptus oil is very well known and used worldwide. Its properties have been widely described in the literature, with particular emphasis on antibacterial and anti-inflammatory properties [52,53]. However, very little information can be found on its antiprotozoal activity. These properties were proved in the following study. Eucalyptus oil showed the same properties as the other tested combinations. It does not change the fact that its effectiveness in combination with acids and metal ions was very high at low concentrations.

Another essential oil analyzed was tea tree essential oil. For years, it has not only been the subject of research, but also an ingredient in many antifungal products [54,55]. It is also undoubtedly effective against *Trichomonas vaginalis* [56]. A human trial showed that a dose of 0.4% tea tree essential oil was as effective as metronidazole treatment [22]. Similar results were obtained in the following work. However, another study showed a 100% antiprotozoal efficacy of tea tree oil at 455 µL/L [57]. Similar results were obtained in the following work. Baldissera et al. showed a much higher effective concentration, 1–2%, in their study [58]. Other tests also showed its high antiprotozoal efficacy, but also high selectivity of the main component of tea tree oil—terpinen-4-ol [59]. The composition containing tea tree oil, together with a mixture of organic acids and metal ions, had the best antiprotozoal properties among the combinations tested. It exhibited a mostly lethal effect, against 50% of the population, at concentrations in the range of 0.001–0.004%.

Cedar essential oil shows very strong antibacterial, antiviral, insecticidal and antiprotozoal properties [60,61,62,63]. Studies have shown that in its pure form, it has efficacy levels of LD_50_ = 0.04–1% and LD_100_ = 0.06–0.25% against analyzed protozoa. Obtained concentrations were very low. Unlike the *C. deodara* species, *Cedrus libani* did not show antiprotozoal properties against *Leishmania major* [64]. Nisha et al. showed similar activity and effective concentration against the adult form of *Setaria digitata* [65]. In their study, Kar et al. showed a significant enhancement of the effect of cedrol, one of the major components of cedar essential oil, as a cedrol-loaded nano-structured lipid carrier [66]. In our studies, the combination of cedar oil with a mixture of acids and metal ions showed very good antiprotozoal properties in the range of 0.001–0.008% against the analyzed protozoa. Compared to the values obtained for cedar essential oil, it is 40 times and more than 30 times more potent, LD_50_ and LD_100_, respectively.

Many studies present the high activity effects of lavender oil against microorganisms [67]. However, there are few reports in the literature on its antiprotozoal activity. The results presented in this paper allow us to conclude that lavender oil, especially in the proposed combinations, show very good antiprotozoal properties. The values for pure lavender oil were promising, while the combinations performed only confirmed this. All analyzed variants of lavender essential oil, mixture of acids and Cu, Mn, Zn ions (LMCu, LMMn and LMZn) showed strong protozoicidal activity at the level of 0.001–0.008%.

This article shows the very good antiprotozoal properties of the innovative combinations. The results obtained in this research are often even a hundred times stronger than those of standard antibiotics.

It is necessary to conduct further tests with using the analyzed mixtures. The obligatory element is undoubtedly the analysis of the toxicity of the combination and its direct influence on organs, as well as accumulation in tissues. However, the mixtures obtained have a very high potential and can be used not only in medicine and pharmacy, but also in the prophylaxis of diseases caused by protozoa, both in humans and animals.

## 4. Materials and Methods

### 4.1. Maintenance of Parasite Cultures and Evaluation of Antiprotozoal Activity

Five organisms representing the taxonomic groups to which the pathogenic protozoa belong were selected for in vitro studies of the antiprotozoal activity of the mixtures:

*Amoeba proteus*—Chaos diffluens—a protozoan of the order *Euamoebida*, belonging to the phylum *Amoebozoa*, living in water.*Paramecium caudatum*—a paramecium representing aquatic ciliates.*Gregarina blattarum*—gregarines were isolated from cockroaches, representing the type *Apicomplexa*, living in the digestive tracts or body cavities of invertebrates.*Euglena gracilis*—a protozoan living in water, representing the flagellates—*Mastigophora*, family *Euglenaceae*.*Pentatrichomonas hominis*—a protozoan that lives in the human colon, representing the *Trichomonadidae*.

The Amoeba, Paramecium and Euglena studied in this work were isolated from the freshwater river in Krosno (river Bado, 49°39’59.8’ N 21°46’28.1’ E, Krosno, Subcarpathian Voivodeship, Poland).

Amoeba proteus was cultivated in the Prescott medium and was fed with ciliates, for example, *Tetrahymena* and *Chilomonas* [68,69]. Paramecium was cultivated in hay infusion [70,71,72]. Euglena was cultivated in solution according to Wu [73]. *Pentatrichomonas hominis* was isolated from stool samples and kept in Pahm solution according to Chomicz et al. [74]

The gregarines were isolated from cockroaches and treated with the mixtures at different concentrations after being placed on a watch glass in Ringer’s solution. Each sample included ten individuals. The isolation of gregarine from cockroaches was carried out according to the method of isolation of gregarines from beetles proposed by J. Moraczewski [75].

Amoeba, Paramecium and Euglena were observed microscopically on a watch glass with viscose wool fibers (to facilitate observation) in a drop of culture water, from which they originated. Different concentrations of the combinations were introduced into the test samples, establishing an LD_50_ dose (50% mortality) and an LD_100_ dose (100% mortality). For determining LD50, the Reed–Muench method was used. In all cases, four-fold replicates of the test were used along with a blank test. The lethal concentration of the substance LD_50_ and LD_100_ within 3 and 5 min was determined.

Identification of individual protozoa was made on the basis of their descriptions and drawings after W. A. Dogiel [76] and J. Hempel-Zawitkowska [35,77].

The obtained mixtures of phytoncides with metals and single phytoncides were dissolved in an aqueous solution of polysorbate 80 (0.05%) before being applied to a watch glass. No biocidal effect of polysorbate 80 was observed at the above concentrations. Chloramphenicol and metronidazole were used as standard substances to control protozoa. Concentration of the antibiotics were 5 mg/mL and the dilutions were prepared from the stock to reach LD_50_ and LD_100._

### 4.2. Essential Oils

Essential oils were ordered from two companies. Eucalyptus and tea tree were purchased from Food Base Kft. (Gödöllő, Hungary), cedar essential oil from Synthite Industries Pvt., Ltd. (Kolenchery, Kerala, India) and lavender essential oil from De Monchy Aromatics Ltd. (Poole, Dorset, UK)

### 4.3. Chemicals and Reagents

Organic acids (acetic acid 99%, propionic acid 99.5% and lactic acid 85%) and other chemical reagents purchased from Sigma-Aldrich (St. Louis, MO, USA) comply with FCC and FG standards. The purity and percentage composition, according to the supplier’s specification, was minimum ≥95%.

### 4.4. Phytoncides Mixture Preparation

Essential oils (100 mL) were added in the same amount to organic acids (100 mL) or mixture of acids (ratio 1:1:1) and 5 g of copper (II) carbonate hydroxide (2.87 g of ions Cu^2+^) or 5 g zinc carbonate (2.61 g of ions Zn^2+^) or 5 g manganese (II) chloride (2.18 g of ions Mn^2+^). The entire mixture was heated until the color changed. The mixture was then allowed to cool to obtain a clear solution (one, two or three phases). After this time, the mixture was filtered through a paper filter. The combination was diluted: 1.5% to 0.001%; after that, the protozoa were placed in each dilution.

### 4.5. GC-MS Analysis

The profile of the essential oils investigated was assessed using the GC-MS technique according to the protocol [78]. Identification of all volatile constituents was based on comparison of experimentally obtained compound’s mass spectra with mass spectra available in the NIST20 database. Additionally, the retention indices (RI) obtained experimentally, calculated using macro [79], were compared with the RI available in the NIST20 database and the data from the literature [80]. Shimadzu software GCMS Postrun Analysis (Shimadzu Company, Kyoto, Japan) and ACD/Spectrus Processor (Advanced Chemistry Development, Inc., Toronto, ON, Canada) were used to process the data. The quantification of identified constituents was performed by calculation based on the amount of added internal standard and expressed as a percentage of integrated peaks’ area. Analysis was performed using the Shimadzu 2020 apparatus (Varian, Walnut Creek, CA, USA) equipped with a Zebron ZB-5 MSI (30 m × 0.25 mm × 0.25 μm) column (Phenomenex, Torrance, CA, USA). The temperature of the GC oven was programmed from 50 °C to 250 °C at a rate of 3.0 °C and kept for 3 min. Scanning was performed from 35 to 550 m/z in electronic impact (EI) at 70 eV and ion source temperature 250 °C. Samples were injected at split ratio 1:10 and gas helium was used as the carrier gas at a flow rate of 1.0 mL/min.

### 4.6. Statistical Analysis

The data, from LD_50_ and LD_100_ evaluation, were subjected to the analysis of variance (ANOVA) using Tukey’s test (*p* < 0.05) using the STATISTICA 13.3 software for Windows (StatSoft, Krakow, Poland).

## 5. Conclusions

In the presented work, a hitherto unused combination of three types of compounds: essential oils, organic acids and metal ions, was used. It has been proved that the proposed combinations show very strong antiprotozoal activity. Studies conducted so far allow to conclude the synergistic effect of these combinations and obtain protozoicidal results much better than standard antibiotics—chloramphenicol or metronidazole. Very high effectiveness against all of the analyzed protozoans was found in the combinations of tea tree, cedar and lavender essential oils, mixture of acids and all of the ions. The LD_50_ and LD_100_ values were in the range 0.001–0.009%. The highest antiprotozoal properties were obtained in the combination with cedar and tea tree essential oils, mixture of acids and manganese or zinc ions. The proposed combinations may find application in eradication of protozoan diseases both in humans and animals. However, further steps should be taken to analyze the antiprotozoal effect on model protozoa such as *Cryptosporidium spp., Leishmania spp.* and *Trypanosoma spp*. as well as toxicological studies of the effective concentrations.

## Figures and Tables

**Table 1 antibiotics-11-00913-t001:** Combinations obtained during the research.

Essential Oil	Acetic Acid (A)			Propionic Acid (P)			Lactic Acid (L)			Mixture of Acids (M)		
	Cu	Mn	Zn	Cu	Mn	Zn	Cu	Mn	Zn	Cu	Mn	Zn
Eucalyptus essential oil (*Eucalyptus globulus* Labill.) (E)	EACu	EAMn	EAZn	EPCu	EPMn	EPZn	ELCu	ELMn	ELZn	EMCu	EMMn	EMZn
Tea tree essential oil (*Melaleuca alternifolia* (Maiden & Betche) Cheel) (T)	TACu	TAMn	TAZn	TPCu	TPMn	TPZn	TLCu	TLMn	TLZn	TMCu	TMMn	TMZn
Cedar essential oil (*Cedrus* sp.) (C)	CACu	CAMn	CAZn	CPCu	CPMn	CPZn	CLCu	CLMn	CLZn	CMCu	CMMn	CMZn
Lavender essential oil (*Lavandula angustifolia* Miller) (L)	LACu	LAMn	LAZn	LPCu	LPMn	LPZn	LLCu	LLMn	LLZn	LMCu	LMMn	LMZn

**Table 2 antibiotics-11-00913-t002:** LD_50_, LD_100_ values of eucalyptus essential oil (*Eucalyptus globulus Labill*.) and the components used in the study.

Protozoa	CH ^a^	M ^b^	Acetic Acid	Propionic Acid	Lactic Acid	Mixture of Acids ^c^	Manganese (II) Chloride Solution ^d^	Copper (II) Carbonate Hydroxide Solution ^e^	Zinc Carbonate Solution ^f^	Catalyst Solution ^g^	Eucalyptus Essential Oil (*Eucalyptus globulus* Labill.)
*Euglena gracilis*	LD_50_: 0.05% LD_100_: 0.09%	LD_50_: n.t LD_100_: n.t	LD_50_: 0.8% LD_100_: 1.1%	LD_50_: 0.5% LD_100_: 1.1%	LD_50_: 0.6% LD_100_: 1.3%	LD_50_: 0.5% LD_100_: 0.9%	LD_50_: 0.5% LD_100_: 0.7%	LD_50_: 0.1% LD_100_: 0.2%	LD_50_: 0.1% LD_100_: 0.3%	LD_50_: 0.5% LD_100_: 0.1%	LD_50_: 0.1% LD_100_: 0.2%
*Gregarina blattarum*	LD_50_: n.t LD_100_: n.t	LD_50_: 0.1% LD_100_: 0.3%	LD_50_: 0.9% LD_100_: 1.1%	LD_50_: 0.9% LD_100_: 1.0%	LD_50_: 1.0% LD_100_: 1.1%	LD_50_: 0.9% LD_100_: 1.0%	LD_50_: 0.4% LD_100_: 0.7%	LD_50_: 0.1% LD_100_: 0.4%	LD_50_: 0.2% LD_100_: 0.4%	LD_50_: 0.7% LD_100_: 0.3%	LD_50_: 0.2% LD_100_: 0.5%
*Amoeba proteus*	LD_50_: 0.07% LD_100_: 0.15%	LD_50_: 0.3% LD_100_: 0.5%	LD_50_: 0.8% LD_100_: 1.0%	LD_50_: 0.6% LD_100_: 1.0%	LD_50_: 0.9% LD_100_: 1.4%	LD_50_: 0.5% LD_100_: 1.0%	LD_50_: 0.5% LD_100_: 1.0%	LD_50_: 0.1% LD_100_: 0.2%	LD_50_: 0.1% LD_100_: 0.2%	LD_50_: 0.5% LD_100_: 1.0%	LD_50_: 0.5% LD_100_: 0.7%
*Paramecium caudatum*	LD_50_: 0.001% LD_100_: 0.006%	LD_50_: n.t LD_100_: n.t	LD_50_: 1.0% LD_100_: 1.3%	LD_50_: 0.8% LD_100_: 1.2%	LD_50_: 1.0% LD_100_: 1.5%	LD_50_: 0.8% LD_100_: 1.2%	LD_50_: 0.8% LD_100_: 1.2%	LD_50_: 0.3% LD_100_: 0.5%	LD_50_: 0.3% LD_100_: 0.5%	LD_50_: 0.8% LD_100_: 1.2%	LD_50_: 0.1% LD_100_: 0.3%
*Pentatrichomonas hominis*	LD_50_: n.t LD_100_: n.t	LD_50_: 0.05% LD_100_: 0.14%	LD_50_: 1.0% LD_100_: 1.5%	LD_50_: 0.8% LD_100_: 1.0%	LD_50_: 0.9% LD_100_: 1.3%	LD_50_: 0.8% LD_100_: 1.0%	LD_50_: 0.9% LD_100_: 1.1%	LD_50_: 0.1% LD_100_: 0.3%	LD_50_: 0.2% LD_100_: 0.4%	LD_50_: 0.9% LD_100_: 1.1%	LD_50_: 0.1% LD_100_: 0.3%

^a^—chloramphenicol, ^b^—metronidazole, ^c^—in rate 1:1:1, ^d^—10% solution, ^e^—10% solution, ^f^—10% solution, ^g^—5% solution, n.t—not tested.

**Table 3 antibiotics-11-00913-t003:** LD_50_, LD_100_ values for the tested mixtures of eucalyptus essential oil (*Eucalyptus globulus* Labill.) (E), organic acids (Acetic acid—A, Propionic acid—P, Lactic acid—L, Mixture of acids—M) and metal ion against selected protozoa.

Protozoa	Eucalyptus Essential Oil (*Eucalyptus globulus* Labill.)											
	Acetic Acid			Propionic Acid			Lactic Acid			Mixture of Acids ^a^		
	Cu ^b^	Mn ^c^	Zn ^d^	Cu ^b^	Mn ^c^	Zn ^d^	Cu ^b^	Mn ^c^	Zn ^d^	Cu ^b^	Mn ^c^	Zn ^d^
	EACu	EAMn	EAZn	EPCu	EPMn	EPZn	ELCu	ELMn	ELZn	EMCu	EMMn	EMZn
*Euglena gracilis* ^1^	LD_50_: 0.04% ± 0.035 ab LD_100_: 0.08% ± 0.068 abc	LD_50_: 0.03% ± 0.032 ab LD_100_: 0.06% ± 0.058 abcd	LD_50_: 0.03% ± 0.032 ab LD_100_: 0.07% ± 0.065 abc	LD_50_: 0.01% ± 0.015 b LD_100_: 0.03% ± 0.025 d	LD_50_: 0.04% ± 0.038 ab LD_100_: 0.08% ± 0.078 a	LD_50_: 0.03% ± 0.028 ab LD_100_: 0.06% ± 0.070 ab	LD_50_: 0.03% ± 0.032 ab LD_100_: 0.07% ± 0.069 abc	LD_50_: 0.01% ± 0.102 a LD_100_: 0.03% ± 0.032 bcd	LD_50_: 0.04% ± 0.038 ab LD_100_: 0.08% ± 0.080 a	LD_50_: 0.03% ± 0.032 ab LD_100_: 0.06% ± 0.088 a	LD_50_: 0.03% ± 0.028 ab LD_100_: 0.07% ± 0.068 abc	LD_50_: 0.01% ± 0.102 a LD_100_: 0.03% ± 0.030 cd
*Gregarina blattarum^1^*	LD_50_: 0.04% ± 0.038 a LD_100_: 0.05% ± 0.045 b	LD_50_: 0.03% ± 0.034 a LD_100_: 0.07% ± 0.065 b	LD_50_: 0.03% ± 0.032 a LD_100_: 0.06% ± 0.060 b	LD_50_: 0.03% ± 0.034 a LD_100_: 0.04% ± 0.038 b	LD_50_: 0.04% ± 0.040 a LD_100_: 0.05% ± 0.052 b	LD_50_: 0.03% ± 0.031 a LD_100_: 0.07% ± 0.068 b	LD_50_: 0.03% ± 0.028 a LD_100_: 0.06% ± 0.062 b	LD_50_: 0.03% ± 0.028 a LD_100_: 0.04% ± 0.042 b	LD_50_: 0.04% ± 0.042 a LD_100_: 0.05% ± 0.049 b	LD_50_: 0.03% ± 0.035 a LD_100_: 0.07% ± 0.248 a	LD_50_: 0.03% ± 0.025 a LD_100_: 0.06% ± 0.056 b	LD_50_: 0.03% ± 0.035 a LD_100_: 0.04% ± 0.041 b
*Amoeba proteus ^1^*	LD_50_: 0.03% ± 0.028 c LD_100_: 0.06% ± 0.062 a	LD_50_: 0.04% ± 0.045 abc LD_100_: 0.08% ± 0.076 a	LD_50_: 0.05% ± 0.052 abc LD_100_: 0.09% ± 0.082 a	LD_50_: 0.07% ± 0.065 ab LD_100_: 0.08% ± 0.075 a	LD_50_: 0.03% ± 0.032 c LD_100_: 0.06% ± 0.062 a	LD_50_: 0.04% ± 0.042 bc LD_100_: 0.08% ± 0.080 a	LD_50_: 0.05% ± 0.049 abc LD_100_: 0.09% ± 0.085 a	LD_50_: 0.07% ± 0.072 a LD_100_: 0.08% ± 0.082 a	LD_50_: 0.03% ± 0.031 c LD_100_: 0.06% ± 0.055 a	LD_50_: 0.04% ± 0.045 abc LD_100_: 0.08% ± 0.082 a	LD_50_: 0.05% ± 0.050 abc LD_100_: 0.09% ± 0.078 a	LD_50_: 0.07% ± 0.070 ab LD_100_: 0.08% ± 0.075 a
*Paramecium caudatum ^1^*	LD_50_: 0.02% ± 0.020 a LD_100_: 0.06% ± 0.062 a	LD_50_: 0.02% ± 0.022 a LD_100_: 0.07% ± 0.070 a	LD_50_: 0.04% ± 0.040 a LD_100_: 0.07% ± 0.072 a	LD_50_: 0.03% ± 0.032 a LD_100_: 0.08% ± 0.075 a	LD_50_: 0.02% ± 0.022 a LD_100_: 0.06% ± 0.062 a	LD_50_: 0.02% ± 0.022 a LD_100_: 0.07% ± 0.072 a	LD_50_: 0.04% ± 0.042 a LD_100_: 0.07% ± 0.071 a	LD_50_: 0.03% ± 0.032 a LD_100_: 0.08% ± 0.075 a	LD_50_: 0.02% ± 0.024 a LD_100_: 0.06% ± 0.065 a	LD_50_: 0.02% ± 0.024 a LD_100_: 0.07% ± 0.069 a	LD_50_: 0.04% ± 0.032 a LD_100_: 0.07% ± 0.065 a	LD_50_: 0.03% ± 0.032 a LD_100_: 0.08% ± 0.078 a
*Pentatrichomonas hominis^1^*	LD_50_: 0.04% ± 0.035 a LD_100_: 0.07% ± 0.072 ab	LD_50_: 0.03% ± 0.032 a LD_100_: 0.05% ± 0.045 b	LD_50_: 0.05% ± 0.050 a LD_100_: 0.09% ± 0.085 ab	LD_50_: 0.03% ± 0.032 a LD_100_: 0.04% ± 0.042 b	LD_50_: 0.04% ± 0.040 a LD_100_: 0.07% ± 0.065 ab	LD_50_: 0.03% ± 0.030 a LD_100_: 0.05% ± 0.050 b	LD_50_: 0.05% ± 0.045 a LD_100_: 0.09% ± 0.085 ab	LD_50_: 0.03% ± 0.035 a LD_100_: 0.04% ± 0.035 b	LD_50_: 0.04% ± 0.040 a LD_100_: 0.07% ± 0.068 ab	LD_50_: 0.03% ± 0.048 a LD_100_: 0.05% ± 0.052 b	LD_50_: 0.05% ± 0.024 a LD_100_: 0.09% ± 0.125 a	LD_50_: 0.03% ± 0.052 a LD_100_: 0.04% ± 0.040 b

^1^ Values followed by the same letter within a row are not significantly different (*p* > 0.05, Tukey’s test), ^a^—in rate 1:1:1, ^b^—10% solution, ^c^—10% solution, ^d^—10% solution, EACu, EAMn, EAZn—Eucalyptus essential oil (E) with acetic acid (A) and Cu, Mn, Zn ions, respectively; EPCu, EPMn, EPZn—Eucalyptus essential oil (E) with propionic acid (P) and Cu, Mn, Zn ions, respectively; ELCu, ELMn, ELZn—Eucalyptus essential oil (E) with lactic acid (L) and Cu, Mn, Zn ions, respectively; EMCu, EMMn, EMZn—Eucalyptus essential oil (E) with mixture of acids (M) and Cu, Mn, Zn ions, respectively.

**Table 4 antibiotics-11-00913-t004:** LD_50_, LD_100_ values of tea tree essential oil (*Melaleuca alternifolia* (Maiden & Betche) Cheel) and the components used in the study.

Protozoa	CH ^a^	M ^b^	Acetic Acid	Propionic Acid	Lactic Acid	Mixture of Acids ^c^	Manganese (II) Chloride Solution ^d^	Copper (II) Carbonate Hydroxide Solution ^e^	Zinc Carbonate Solution ^f^	Catalyst Solution ^g^	Tea Tree Essential Oil (*Melaleuca alternifolia* (Maiden & Betche) Cheel *)*
*Euglena gracilis*	LD_50_: 0.05% LD_100_: 0.09%	LD_50_: n.t LD_100_: n.t	LD_50_: 0.8% LD_100_: 1.1%	LD_50_: 0.5% LD_100_: 1.1%	LD_50_: 0.6% LD_100_: 1.3%	LD_50_: 0.5% LD_100_: 0.9%	LD_50_: 0.5% LD_100_: 0.7%	LD_50_: 0.1% LD_100_: 0.2%	LD_50_: 0.1% LD_100_: 0.3%	LD_50_: 0.5% LD_100_: 0.1%	LD_50_: 0.05% LD_100_: 0.1%
*Gregarina blattarum*	LD_50_: n.t LD_100_: n.t	LD_50_: 0.1% LD_100_: 0.3%	LD_50_: 0.9% LD_100_: 1.1%	LD_50_: 0.9% LD_100_: 1.0%	LD_50_: 1.0% LD_100_: 1.1%	LD_50_: 0.9% LD_100_: 1.0%	LD_50_: 0.4% LD_100_: 0.7%	LD_50_: 0.1% LD_100_: 0.4%	LD_50_: 0.2% LD_100_: 0.4%	LD_50_: 0.7% LD_100_: 0.3%	LD_50_: 0.25% LD_100_: 0.3%
*Amoeba proteus*	LD_50_: 0.07% LD_100_: 0.15%	LD_50_: 0.3% LD_100_: 0.5%	LD_50_: 0.8% LD_100_: 1.0%	LD_50_: 0.6% LD_100_: 1.0%	LD_50_: 0.9% LD_100_: 1.4%	LD_50_: 0.5% LD_100_: 1.0%	LD_50_: 0.5% LD_100_: 1.0%	LD_50_: 0.1% LD_100_: 0.2%	LD_50_: 0.1% LD_100_: 0.2%	LD_50_: 0.5% LD_100_: 1.0%	LD_50_: 0.3% LD_100_: 0.5%
*Paramecium caudatum*	LD_50_: 0.001% LD_100_: 0.006%	LD_50_: n.t LD_100_: n.t	LD_50_: 1.0% LD_100_: 1.3%	LD_50_: 0.8% LD_100_: 1.2%	LD_50_: 1.0% LD_100_: 1.5%	LD_50_: 0.8% LD_100_: 1.2%	LD_50_: 0.8% LD_100_: 1.2%	LD_50_: 0.3% LD_100_: 0.5%	LD_50_: 0.3% LD_100_: 0.5%	LD_50_: 0.8% LD_100_: 1.2%	LD_50_: 0.2% LD_100_: 0.25%
*Pentatrichomonas hominis*	LD_50_: n.t LD_100_: n.t	LD_50_: 0.05% LD_100_: 0.14%	LD_50_: 1.0% LD_100_: 1.5%	LD_50_: 0.8% LD_100_: 1.0%	LD_50_: 0.9% LD_100_: 1.3%	LD_50_: 0.8% LD_100_: 1.0%	LD_50_: 0.9% LD_100_: 1.1%	LD_50_: 0.1% LD_100_: 0.3%	LD_50_: 0.2% LD_100_: 0.4%	LD_50_: 0.9% LD_100_: 1.1%	LD_50_: 0.08% LD_100_: 0.1%

^a^—chloramphenicol, ^b^—metronidazole, ^c^—in rate 1:1:1, ^d^—10% solution, ^e^—10% solution, ^f^—10% solution, ^g^—5% solution, n.t—not tested.

**Table 5 antibiotics-11-00913-t005:** LD_50_, LD_100_ values of tea tree essential oil (*Melaleuca alternifolia* (Maiden & Betche) Cheel) (T), organic acids (Acetic acid—A, Propionic acid—P, Lactic acid—L, Mixture of acids—M) and metal ion against selected protozoa.

Protozoa	Tea Tree Essential Oil (*Melaleuca alternifolia* (Maiden & Betche) Cheel)											
	Acetic Acid			Propionic Acid			Lactic Acid			Mixture of Acids ^a^		
	Cu ^b^	Mn ^c^	Zn ^d^	Cu ^b^	Mn ^c^	Zn ^d^	Cu ^b^	Mn ^c^	Zn ^d^	Cu ^b^	Mn ^c^	Zn ^d^
	TACu	TAMn	TAZn	TPCu	TPMn	TPZn	TLCu	TLMn	TLZn	TMCu	TMMn	TMZn
*Euglena gracilis* ^1^	LD_50_: 0.03% ± 0.032 a LD_100_: 0.06% ± 0.059 ab	LD_50_: 0.04% ± 0.035 a LD_100_: 0.07% ± 0.065 a	LD_50_: 0.03% ± LD_100_: 0.05% ± 0.052 abc	LD_50_: 0.03% ± 0.026 a LD_100_: 0.04% ± 0.035 bcd	LD_50_: 0.01% ± 0.0134 ab LD_100_: 0.02% ± 0.018 de	LD_50_: 0.02% ± 0.020 ab LD_100_: 0.03% ± 0.028 cde	LD_50_: 0.02% ± 0.021 ab LD_100_: 0.04% ± 0.038 bcd	LD_50_: 0.02% ± 0.022 ab LD_100_: 0.04% ± 0.042 abcd	LD_50_: 0.02% ± 0.022 ab LD_100_: 0.05% ± 0.052 abc	LD_50_: 0.001% ± 0.001 b LD_100_: 0.003% ± 0.002 e	LD_50_: 0.001% ± 0.001 b LD_100_: 0.002% ± 0.002 e	LD_50_: 0.003% ± 0.002 b LD_100_: 0.004% ± 0.004 e
*Gregarina blattarum* ^1^	LD_50_: 0.04% ± 0.038 b LD_100_: 0.07% ± 0.065 b	LD_50_: 0.04% ± 0.036 b LD_100_: 0.05% ± 0.052 bcd	LD_50_: 0.04% ± 0.035 b LD_100_: 0.06% ± 0.060 b	LD_50_: 0.04% ± 0.036 b LD_100_: 0.06% ± 0.058 bc	LD_50_: 0.05% ± 0.048 b LD_100_: 0.08% ± 0.078 b	LD_50_: 0.05% ± 0.051 b LD_100_: 0.07% ± 0.068 b	LD_50_: 0.03% ± 0.031 b LD_100_: 0.04% ± 0.038 bcd	LD_50_: 0.05% ± 0.049 b LD_100_: 0.07% ± 0.065 b	LD_50_: 0.08% ± 0.075 a LD_100_: 0.15% ± 0.138 a	LD_50_: 0.004% ± 0.004 c LD_100_: 0.006% ± 0.006 d	LD_50_: 0.004% ± 0.004 c LD_100_: 0.007% ± 0.007 cd	LD_50_: 0.003% ± 0.002 c LD_100_: 0.005% ± 0.004 d
*Amoeba proteus* ^1^	LD_50_: 0.04% ± 0.038 bc LD_100_: 0.06% ± 0.062 bc	LD_50_: 0.03% ± 0.035 cd LD_100_: 0.06% ± 0.06 4 bc	LD_50_: 0.04% ± 0.042 bc LD_100_: 0.07% ± 0.072 abc	LD_50_: 0.04% ± 0.042 bc LD_100_: 0.07% ± 0.072 abc	LD_50_: 0.01% ± 0.014 de LD_100_: 0.01% ± 0.014 de	LD_50_: 0.07% ± 0.072 a LD_100_: 0.09% ± 0.092 a	LD_50_: 0.06% ± 0.058 ab LD_100_: 0.08% ± 0.082 ab	LD_50_: 0.01% ± 0.012 e LD_100_: 0.03% ± 0.028 d	LD_50_: 0.01% ± 0.016 de LD_100_: 0.05% ± 0.052 c	LD_50_: 0.002% ± 0.002 e LD_100_: 0.003% ± 0.004 de	LD_50_: 0.002% ± 0.002 e LD_100_: 0.004% ± 0.003 e	LD_50_: 0.001% ± 0.001 e LD_100_: 0.002% ± 0.002 e
*Paramecium caudatum* ^1^	LD_50_: 0.02% ± 0.022 abcd LD_100_: 0.05% ± 0.052 bcd	LD_50_: 0.02% ± 0.020 abcd LD_100_: 0.05% ± 0.045 cde	LD_50_: 0.03% ± 0.032 abcd LD_100_: 0.06% ± 0.062 abc	LD_50_: 0.05% ± 0.045 ab LD_100_: 0.08% ± 0.075 ab	LD_50_: 0.01% ± 0.012 bcd LD_100_: 0.02% ± 0.022 ef	LD_50_: 0.05% ± 0.050 a LD_100_: 0.09% ± 0.085 a	LD_50_: 0.05% ± 0.049 a LD_100_: 0.07% ± 0.068 abc	LD_50_: 0.03% ± 0.035 abc LD_100_: 0.04% ± 0.042 cde	LD_50_: 0.02% ± 0.024 abcd LD_100_: 0.03% ± 0.032 de	LD_50_: 0.002% ± 0.002 cd LD_100_: 0.005% ± 0.005 f	LD_50_: 0.001% ± 0.001 d LD_100_: 0.002% ± 0.002 f	LD_50_: 0.003% ± 0.003 cd LD_100_: 0.005% ± 0.005 f
*Pentatrichomonas hominis* ^1^	LD_50_: 0.09% ± 0.085 a LD_100_: 0.35% ± 0.362 a	LD_50_: 0.08% ± 0.075 a LD_100_: 0.2% ± 0.238 a	LD_50_: 0.07% ± 0.065 ab LD_100_: 0.25% ± 0.238 a	LD_50_: 0.03% ± 0.032 cd LD_100_: 0.05% ± 0.045 b	LD_50_: 0.01% ± 0.012 de LD_100_: 0.04% ± 0.035 b	LD_50_: 0.07% ± 0.065 ab LD_100_: 0.09% ± 0.088 b	LD_50_: 0.025% ± 0.025 cde LD_100_: 0.045% ± 0.046 b	LD_50_: 0.05% ± 0.045 bc LD_100_: 0.07% ± 0.072 b	LD_50_: 0.02% ± 0.020 cde LD_100_: 0.05% ± 0.045 b	LD_50_: 0.004% ± 0.004 e LD_100_: 0.004% ± 0.004 b	LD_50_: 0.007% ± 0.007 de LD_100_: 0.009% ± 0.008 b	LD_50_: 0.002% ± 0.002 e LD_100_: 0.004% ± 0.002 b

^1^ Values followed by the same letter within a row are not significantly different (*p* > 0.05, Tukey’s test), ^a^—in rate 1:1:1, ^b^—10% solution, ^c^—10% solution, ^d^—10% solution, TACu, TAMn, TAZn—Tea tree essential oil (T) with acetic acid (A) and Cu, Mn, Zn ions, respectively; TPCu, TPMn, TPZn—Tea tree essential oil (T) with propionic acid (P) and Cu, Mn, Zn ions, respectively; TLCu, TLMn, TLZn—Tea tree essential oil (T) with lactic acid (L) and Cu, Mn, Zn ions, respectively; TMCu, TMMn, TMZn—Tea tree essential oil (T) with mixture of acids (M) and Cu, Mn, Zn ions, respectively.

**Table 6 antibiotics-11-00913-t006:** LD_50_, LD_100_ values of cedar essential oil (*Cedrus sp*.) and the components used in the study.

Protozoa	CH ^a^	M ^b^	Acetic Acid	Propionic Acid	Lactic Acid	Mixture of Acids ^c^	Manganese (II) Chloride Solution ^d^	Copper (II) Carbonate Hydroxide Solution ^e^	Zinc Carbonate Solution ^f^	Catalyst Solution ^g^	Cedar Essential Oil (*Cedrus* sp.)
*Euglena gracilis*	LD_50_: 0.05% LD_100_: 0.09%	LD_50_: n.t LD_100_: n.t	LD_50_: 0.8% LD_100_: 1.1%	LD_50_: 0.5% LD_100_: 1.1%	LD_50_: 0.6% LD_100_: 1.3%	LD_50_: 0.5% LD_100_: 0.9%	LD_50_: 0.5% LD_100_: 0.7%	LD_50_: 0.1% LD_100_: 0.2%	LD_50_: 0.1% LD_100_: 0.3%	LD_50_: 0.5% LD_100_: 0.1%	LD_50_: 0.7% LD_100_: 0.9%
*Gregarina blattarum*	LD_50_: n.t LD_100_: n.t	LD_50_: 0.1% LD_100_: 0.3%	LD_50_: 0.9% LD_100_: 1.1%	LD_50_: 0.9% LD_100_: 1.0%	LD_50_: 1.0% LD_100_: 1.1%	LD_50_: 0.9% LD_100_: 1.0%	LD_50_: 0.4% LD_100_: 0.7%	LD_50_: 0.1% LD_100_: 0.4%	LD_50_: 0.2% LD_100_: 0.4%	LD_50_: 0.7% LD_100_: 0.3%	LD_50_: 0.7% LD_100_: 0.9%
*Amoeba proteus*	LD_50_: 0.07% LD_100_: 0.15%	LD_50_: 0.3% LD_100_: 0.5%	LD_50_: 0.8% LD_100_: 1.0%	LD_50_: 0.6% LD_100_: 1.0%	LD_50_: 0.9% LD_100_: 1.4%	LD_50_: 0.5% LD_100_: 1.0%	LD_50_: 0.5% LD_100_: 1.0%	LD_50_: 0.1% LD_100_: 0.2%	LD_50_: 0.1% LD_100_: 0.2%	LD_50_: 0.5% LD_100_: 1.0%	LD_50_: 0.4% LD_100_: 0.6%
*Paramecium caudatum*	LD_50_: 0.001% LD_100_: 0.006%	LD_50_: n.t LD_100_: n.t	LD_50_: 1.0% LD_100_: 1.3%	LD_50_: 0.8% LD_100_: 1.2%	LD_50_: 1.0% LD_100_: 1.5%	LD_50_: 0.8% LD_100_: 1.2%	LD_50_: 0.8% LD_100_: 1.2%	LD_50_: 0.3% LD_100_: 0.5%	LD_50_: 0.3% LD_100_: 0.5%	LD_50_: 0.8% LD_100_: 1.2%	LD_50_: 0.1% LD_100_: 0.25%
*Pentatrichomonas hominis*	LD_50_: n.t LD_100_: n.t	LD_50_: 0.05% LD_100_: 0.14%	LD_50_: 1.0% LD_100_: 1.5%	LD_50_: 0.8% LD_100_: 1.0%	LD_50_: 0.9% LD_100_: 1.3%	LD_50_: 0.8% LD_100_: 1.0%	LD_50_: 0.9% LD_100_: 1.1%	LD_50_: 0.1% LD_100_: 0.3%	LD_50_: 0.2% LD_100_: 0.4%	LD_50_: 0.9% LD_100_: 1.1%	LD_50_: 0.1% LD_100_: 0.2%

^a^—chloramphenicol, ^b^—metronidazole, ^c^—in rate 1:1:1, ^d^—10% solution, ^e^—10% solution, ^f^—10% solution, ^g^—5% solution, n.t—not tested.

**Table 7 antibiotics-11-00913-t007:** LD_50_, LD_100_ values of cedar essential oil (*Cedrus sp*.) (C), organic acids (Acetic acid—A, Propionic acid—P, Lactic acid—L, Mixture of acids—M) and metal ion against selected protozoa.

Protozoa	Cedar Essential Oil (*Cedrus* sp.)											
	Acetic Acid			Propionic Acid			Lactic Acid			Mixture of Acids ^a^		
	Cu ^b^	Mn ^c^	Zn ^d^	Cu ^b^	Mn ^c^	Zn ^d^	Cu ^b^	Mn ^c^	Zn ^d^	Cu ^b^	Mn ^c^	Zn ^d^
	CACu	CAMn	CAZn	CPCu	CPMn	CPZn	CLCu	CLMn	CLZn	CMCu	CMMn	CMZn
*Euglena gracilis* ^1^	LD_50_: 0.04% ± 0.035 a LD_100_: 0.08% ± 0.079 a	LD_50_: 0.03% ± 0.033 a LD_100_: 0.07% ± 0.065 abc	LD_50_: 0.04% ± 0.035 a LD_100_: 0.07% ± 0.068 ab	LD_50_: 0.03% ± 0.026 a LD_100_: 0.05% ± 0.048 bcd	LD_50_: 0.01% ± 0.014 ab LD_100_: 0.02% ± 0.018 fg	LD_50_: 0.02% ± 0.020 ab LD_100_: 0.03% ± 0.028 def	LD_50_: 0.01% ± 0.014 ab LD_100_: 0.02% ± 0.023 efg	LD_50_: 0.02% ± 0.023 ab LD_100_: 0.04% ± 0.043 cde	LD_50_: 0.03% ± 0.025 a LD_100_: 0.06% ± 0.055 bc	LD_50_: 0.001% ± 0.001 b LD_100_: 0.002% ± 0.002 g	LD_50_: 0.002% ± 0.002 b LD_100_: 0.004% ± 0.004 g	LD_50_: 0.003% ± 0.003 b LD_100_: 0.006% ± 0.007 fg
*Gregarina blattarum* ^1^	LD_50_: 0.03% ± 0.032 a LD_100_: 0.06% ± 0.062 abc	LD_50_: 0.02% ± 0.022 ab LD_100_: 0.05% ± 0.052 bc	LD_50_: 0.04% ± 0.035 a LD_100_: 0.08% ± 0.082 a	LD_50_: 0.03% ± 0.034 a LD_100_: 0.04% ± 0.042 c	LD_50_: 0.03% ± 0.033 a LD_100_: 0.07% ± 0.070 ab	LD_50_: 0.03% ± 0.031 a LD_100_: 0.05% ± 0.052 bc	LD_50_: 0.03% ± 0.031 a LD_100_: 0.04% ± 0.038 c	LD_50_: 0.02% ± 0.022 ab LD_100_: 0.05% ± 0.049 bc	LD_50_: 0.04% ± 0.043 a LD_100_: 0.05% ± 0.045 bc	LD_50_: 0.003% ± 0.003 b LD_100_: 0.004% ± 0.004 d	LD_50_: 0.002% ± 0.002 b LD_100_: 0.005% ± 0.005 d	LD_50_: 0.004% ± 0.004 b LD_100_: 0.005% ± 0.005 d
*Amoeba proteus* ^1^	LD_50_: 0.04% ± 0.038 ab LD_100_: 0.05% ± 0.052 b	LD_50_: 0.05% ± 0.052 a LD_100_: 0.08% ± 0.076 ab	LD_50_: 0.06% ± 0.062 a LD_100_: 0.01% ± 0.014 c	LD_50_: 0.05% ± 0.045 ab LD_100_: 0.07% ± 0.072 ab	LD_50_: 0.05% ± 0.045 ab LD_100_: 0.08% ± 0.078 ab	LD_50_: 0.04% ± 0.038 ab LD_100_: 0.06% ± 0.058 ab	LD_50_: 0.06% ± 0.058 a LD_100_: 0.08% ± 0.082 a	LD_50_: 0.05% ± 0.045 ab LD_100_: 0.07% ± 0.072 ab	LD_50_: 0.02% ± 0.019 bc LD_100_: 0.05% ± 0.052 b	LD_50_: 0.006% ± 0.006 c LD_100_: 0.008% ± 0.008 c	LD_50_: 0.005% ± 0.005 c LD_100_: 0.007% ± 0.007 c	LD_50_: 0.002% ± 0.002 c LD_100_: 0.005% ± 0.005 c
*Paramecium caudatum* ^1^	LD_50_: 0.03% ± 0.025 ab LD_100_: 0.08% ± 0.082 a	LD_50_: 0.04% ± 0.040 a LD_100_: 0.07% ± 0.072 a	LD_50_: 0.05% ± 0.048 a LD_100_: 0.08% ± 0.075 a	LD_50_: 0.04% ± 0.042 a LD_100_: 0.08% ± 0.075 a	LD_50_: 0.04% ± 0.042 a LD_100_: 0.06% ± 0.060 abc	LD_50_: 0.04% ± 0.040 a LD_100_: 0.07% ± 0.068 ab	LD_50_: 0.03% ± 0.030 ab LD_100_: 0.04% ± 0.042 bc	LD_50_: 0.03% ± 0.035 a LD_100_: 0.08% ± 0.078 a	LD_50_: 0.02% ± 0.024 ab LD_100_: 0.04% ± 0.035 cd	LD_50_: 0.003% ± 0.002 b LD_100_: 0.004% ± 0.004 e	LD_50_: 0.003% ± 0.002 b LD_100_: 0.008% ± 0.008 de	LD_50_: 0.002% ± 0.002 b LD_100_: 0.004% ± 0.004 e
*Pentatrichomonas hominis* ^1^	LD_50_: 0.09% ± 0.085 a LD_100_: 0.15% ± 0.150 a	LD_50_: 0.02% ± 0.020 bc LD_100_: 0.04% ± 0.042 bcde	LD_50_: 0.04% ± 0.040 b LD_100_: 0.07% ± 0.072 b	LD_50_: 0.04% ± 0.035 b LD_100_: 0.06% ± 0.058 bc	LD_50_: 0.03% ± 0.025 bc LD_100_: 0.04% ± 0.040 bcde	LD_50_: 0.03% ± 0.025 bc LD_100_: 0.05% ± 0.048 bcd	LD_50_: 0.03% ± 0.032 b LD_100_: 0.04% ± 0.042 bcde	LD_50_: 0.02% ± 0.022 bc LD_100_: 0.03% ± 0.025 cde	LD_50_: 0.025% ± 0.026 bc LD_100_: 0.07% ± 0.068 b	LD_50_: 0.003% ± 0.002 c LD_100_: 0.004% ± 0.004 e	LD_50_: 0.002% ± 0.002 c LD_100_: 0.003% ± 0.002 e	LD_50_: 0.002% ± 0.002 c LD_100_: 0.007% ± 0.006 de

^1^ Values followed by the same letter within a row are not significantly different (*p* > 0.05, Tukey’s test), ^a^—in rate 1:1:1, ^b^—10% solution, ^c^—10% solution, ^d^—10% solution; CACu, CAMn, CAZn—Cedar essential oil (C) with acetic acid (A) and Cu, Mn, Zn ions, respectively; CPCu, CPMn, CPZn—Cedar essential oil (C) with propionic acid (P) and Cu, Mn, Zn ions, respectively; CLCu, CLMn, CLZn—Cedar essential oil (C) with lactic acid (L) and Cu, Mn, Zn ions, respectively; CMCu, CMMn, CMZn—Cedar essential oil (C) with mixture of acids (M) and Cu, Mn, Zn ions, respectively.

**Table 8 antibiotics-11-00913-t008:** LD_50_, LD_100_ values of lavender essential oil (*Lavandula angustifolia* Miller) and the components used in the study.

Protozoa	CH ^a^	M ^b^	Acetic Acid	Propionic Acid	Lactic Acid	Mixture of Acids ^c^	Manganese (II) Chloride Solution ^d^	Copper (II) Carbonate Hydroxide Solution ^e^	Zinc Carbonate Solution ^f^	Catalyst Solution ^g^	Lavender Essential Oil (*Lavandula angustifolia* Miller)
*Euglena gracilis*	LD_50_: 0.05% LD_100_: 0.09%	LD_50_: n.t LD_100_: n.t	LD_50_: 0.8% LD_100_: 1.1%	LD_50_: 0.5% LD_100_: 1.1%	LD_50_: 0.6% LD_100_: 1.3%	LD_50_: 0.5% LD_100_: 0.9%	LD_50_: 0.5% LD_100_: 0.7%	LD_50_: 0.1% LD_100_: 0.2%	LD_50_: 0.1% LD_100_: 0.3%	LD_50_: 0.5% LD_100_: 0.1%	LD_50_: 0.8% LD_100_: 0.9%
*Gregarina blattarum*	LD_50_: n.t LD_100_: n.t	LD_50_: 0.1% LD_100_: 0.3%	LD_50_: 0.9% LD_100_: 1.1%	LD_50_: 0.9% LD_100_: 1.0%	LD_50_: 1.0% LD_100_: 1.1%	LD_50_: 0.9% LD_100_: 1.0%	LD_50_: 0.4% LD_100_: 0.7%	LD_50_: 0.1% LD_100_: 0.4%	LD_50_: 0.2% LD_100_: 0.4%	LD_50_: 0.7% LD_100_: 0.3%	LD_50_: 0.7% LD_100_: 0.8%
*Amoeba proteus*	LD_50_: 0.07% LD_100_: 0.15%	LD_50_: 0.3% LD_100_: 0.5%	LD_50_: 0.8% LD_100_: 1.0%	LD_50_: 0.6% LD_100_: 1.0%	LD_50_: 0.9% LD_100_: 1.4%	LD_50_: 0.5% LD_100_: 1.0%	LD_50_: 0.5% LD_100_: 1.0%	LD_50_: 0.1% LD_100_: 0.2%	LD_50_: 0.1% LD_100_: 0.2%	LD_50_: 0.5% LD_100_: 1.0%	LD_50_: 0.5% LD_100_: 0.8%
*Paramecium caudatum*	LD_50_: 0.001% LD_100_: 0.006%	LD_50_: n.t LD_100_: n.t	LD_50_: 1.0% LD_100_: 1.3%	LD_50_: 0.8% LD_100_: 1.2%	LD_50_: 1.0% LD_100_: 1.5%	LD_50_: 0.8% LD_100_: 1.2%	LD_50_: 0.8% LD_100_: 1.2%	LD_50_: 0.3% LD_100_: 0.5%	LD_50_: 0.3% LD_100_: 0.5%	LD_50_: 0.8% LD_100_: 1.2%	LD_50_: 0.1% LD_100_: 0.25%
*Pentatrichomonas hominis*	LD_50_: n.t LD_100_: n.t	LD_50_: 0.05% LD_100_: 0.14%	LD_50_: 1.0% LD_100_: 1.5%	LD_50_: 0.8% LD_100_: 1.0%	LD_50_: 0.9% LD_100_: 1.3%	LD_50_: 0.8% LD_100_: 1.0%	LD_50_: 0.9% LD_100_: 1.1%	LD_50_: 0.1% LD_100_: 0.3%	LD_50_: 0.2% LD_100_: 0.4%	LD_50_: 0.9% LD_100_: 1.1%	LD_50_: 0.2% LD_100_: 0.4%

^a^—chloramphenicol, ^b^—metronidazole, ^c^—in rate 1:1:1, ^d^—10% solution, ^e^—10% solution, ^f^—10% solution, ^g^—5% solution, n.t–not tested.

**Table 9 antibiotics-11-00913-t009:** LD_50_, LD_100_ values of lavender essential oil (*Lavandula angustifolia* Miller) (L), organic acids (Acetic acid—A, Propionic acid—P, Lactic acid—L, Mixture of acids—M) and metal ion against selected protozoal.

Protozoa	Lavender Essential Oil (*Lavandula angustifolia* Miller)											
	Acetic Acid			Propionic Acid			Lactic Acid			Mixture of Acids ^a^		
	Cu ^b^	Mn ^c^	Zn ^d^	Cu ^b^	Mn ^c^	Zn ^d^	Cu ^b^	Mn ^c^	Zn ^d^	Cu ^b^	Mn ^c^	Zn ^d^
	LACu	LAMn	LAZn	LPCu	LPMn	LPZn	LLCu	LLMn	LLZn	LMCu	LMMn	LMZn
*Euglena gracilis* ^1^	LD_50_: 0.04% ± 0.035 ab LD_100_: 0.06% ± ± 0.059 ab	LD_50_: 0.02% ± 0.024 bcd LD_100_: 0.04% ± 0.035 cd	LD_50_: 0.05% ± 0.045 a LD_100_: 0.08% ± 0.078 a	LD_50_: 0.03% ± 0.026 abc LD_100_: 0.045% ± 0.046 bcd	LD_50_: 0.01% ± 0.014 cde LD_100_: 0.03% ± 0.025 de	LD_50_: 0.03% ± 0.028 abc LD_100_: 0.05% ± 0.050 bc	LD_50_: 0.02% ± 0.016 bcde LD_100_: 0.04% ± 0.042 bcd	LD_50_: 0.03% ± 0.025 abc LD_100_: 0.05% ± 0.045 bcd	LD_50_: 0.03% ± 0.025 abc LD_100_: 0.05% ± 0.052 bc	LD_50_: 0.002% ± 0.002 e LD_100_: 0.004% ± 0.004 e	LD_50_: 0.001% ± 0.001 e LD_100_: 0.005% ± 0.004 e	LD_50_: 0.004% ± 0.004 de LD_100_: 0.006% ± 0.006 e
*Gregarina blattarum* ^1^	LD_50_: 0.03% ± 0.032 abc LD_100_: 0.06% ± 0.062 ab	LD_50_: 0.01% ± 0.011 cd LD_100_: 0.04% ± 0.040 b	LD_50_: 0.03% ± 0.032 abc LD_100_: 0.08% ± 0.082 a	LD_50_: 0.03% ± 0.034 ab LD_100_: 0.045% ± 0.044 b	LD_50_: 0.03% ± 0.032 abc LD_100_: 0.06% ± 0.060 ab	LD_50_: 0.04% ± 0.036 ab LD_100_: 0.05% ± 0.052 b	LD_50_: 0.05% ± 0.049 a LD_100_: 0.07% ± 0.065 ab	LD_50_: 0.02% ± 0.022 bcd LD_100_: 0.04% ± 0.042 b	LD_50_: 0.05% ± 0.045 a LD_100_: 0.09% ± 0.085 a	LD_50_: 0.003% ± 0.003 d LD_100_: 0.004% ± 0.004 c	LD_50_: 0.002% ± 0.002 d LD_100_: 0.005% ± 0.005 c	LD_50_: 0.004% ± 0.004 d LD_100_: 0.006% ± 0.006 c
*Amoeba proteus* ^1^	LD_50_: 0.05% ± 0.052 ab LD_100_: 0.06% ± 0.055 b	LD_50_: 0.05% ± 0.052 ab LD_100_: 0.07% ± 0.074 ab	LD_50_: 0.06% ± 0.062 a LD_100_: 0.09% ± 0.085 a	LD_50_: 0.04% ± 0.042 abc LD_100_: 0.06% ± 0.060 ab	LD_50_: 0.05% ± 0.045 ab LD_100_: 0.07% ± 0.070 ab	LD_50_: 0.04% ± 0.038 abc LD_100_: 0.07% ± 0.068 ab	LD_50_: 0.02% ± 0.016 cd LD_100_: 0.05% ± 0.049 b	LD_50_: 0.04% ± 0.042 abc LD_100_: 0.07% ± 0.072 ab	LD_50_: 0.03% ± 0.026 bcd LD_100_: 0.06% ± 0.055 b	LD_50_: 0.003% ± 0.003 d LD_100_: 0.004% ± 0.004 c	LD_50_: 0.002% ± 0.002 d LD_100_: 0.004% ± 0.004 c	LD_50_: 0.003% ± 0.002 d LD_100_: 0.005% ± 0.005 c
*Paramecium caudatum* ^1^	LD_50_: 0.05% ± 0.050 a LD_100_: 0.08% ± 0.082 a	LD_50_: 0.05% ± 0.048 ab LD_100_: 0.08% ± 0.075 a	LD_50_: 0.03% ± 0.032 abc LD_100_: 0.07% ± 0.072 a	LD_50_: 0.05% ± 0.045 ab LD_100_: 0.08% ± 0.075 a	LD_50_: 0.03% ± 0.032 abc LD_100_: 0.06% ± 0.060 ab	LD_50_: 0.03% ± 0.030 bc LD_100_: 0.08% ± 0.080 a	LD_50_: 0.02% ± 0.018 cd LD_100_: 0.04% ± 0.042 b	LD_50_: 0.04% ± 0.039 ab LD_100_: 0.07% ± 0.072 a	LD_50_: 0.04% ± 0.035 abc LD_100_: 0.06% ± 0.062 ab	LD_50_: 0.006% ± 0.006 d LD_100_: 0.008% ± 0.008 c	LD_50_: 0.005% ± 0.005 d LD_100_: 0.008% ± 0.008 c	LD_50_: 0.004% ± 0.004 d LD_100_: 0.006% ± 0.006 c
*Pentatrichomonas hominis* ^1^	LD_50_: 0.07% ± 0.070 ab LD_100_: 0.1% ± 0.125 a	LD_50_: 0.07% ± 0.072 ab LD_100_: 0.09% ± 0.085 bc	LD_50_: 0.06% ± 0.040 cd LD_100_: 0.08% ± 0.075 bc	LD_50_: 0.08% ± 0.075 ab LD_100_: 0.1% ± 0.098 abc	LD_50_: 0.06% ± 0.025 de LD_100_: 0.09% ± 0.092 abc	LD_50_: 0.08% ± 0.080 a LD_100_: 0.1% ± 0.110 ab	LD_50_: 0.05% ± 0.048 bcd LD_100_: 0.07% ± 0.072 c	LD_50_: 0.08% ± 0.075 ab LD_100_: 0.1% ± 0.110 ab	LD_50_: 0.055% ± 0.054 abc LD_100_: 0.09% ± 0.088 bc	LD_50_: 0.003% ± 0.002 e LD_100_: 0.005% ± 0.005 d	LD_50_: 0.003% ± 0.002 e LD_100_: 0.007% ± 0.006 d	LD_50_: 0.004% ± 0.004 e LD_100_: 0.008% ± 0.008 d

^1^ Values followed by the same letter within a row are not significantly different (*p* > 0.05, Tukey’s test), ^a^—in rate 1:1:1, ^b^—10% solution, ^c^—10% solution, ^d^—10% solution; LACu, LAMn, LAZn—Lavender essential oil (L) with acetic acid (A) and Cu, Mn, Zn ions, respectively; LPCu, LPMn, LPZn—Lavender essential oil (L) with propionic acid (P) and Cu, Mn, Zn ions, respectively; LLCu, LLMn, LLZn—Lavender essential oil (L) with lactic acid (L) and Cu, Mn, Zn ions, respectively; LMCu, LMMn, LMZn—Lavender essential oil (L) with mixture of acids (M) and Cu, Mn, Zn ions, respectively.

**Table 10 antibiotics-11-00913-t010:** The most efficient combination against tested protozoan.

Protozoa	LD_50_	LD_100_:
*Euglena gracilis*	TMCu, TMMN, CMCu, LMMn	TMMn, CMCu
*Gregarina blattarum*	CMMn, LMMn	CMCu, LMCu
*Amoeba proteus*	TMZn	TMZn
*Paramecium caudatum*	TMMn	TMMn
*Pentatrichomonas hominis*	TMZn, CMMn, CMZn	CMMn

## Data Availability

The data presented in this study are available in the Appendix A.

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
