# Peer review of "Novel Formula of Antiprotozoal Mixtures"

_antibiotics, 2022, doi:10.3390/antibiotics11070913_

Round 1

Reviewer 1 Report

I have evaluated the manuscript (Antibiotics-1777636) titled “Novel formula of antiprotozoal mixtures” by Iwiński and coworkers, and extensive research was done to find out a highly potent mixture of antiprotozoal activity of a mixture of essential oils to combat antibiotics resistance pathogenic microorganisms. All standard methods were used for this research. I found this article interesting for the readers and followed the journal Antibiotics’ scope. I don’t have any major comments as this article have enough data however, the author needs to work on the presentation of results, needs to elaborate on the discussion.

I would recommend the article could be published in Antibiotics after minor corrections. 

The author needs to address the following comments/corrections.

 1.     The author could have used an abbreviation with the full name when it first appears in the manuscript.

2.     Introduction should be short, concise, and to the point. No need to discuss and elaborate well know topics.

3.     All the tables should include footnotes.

4.     The author could have discussed the rationale behind the composition of the mixtures.

5.     Supplementary documents should include title, author’s name, and table of content. All the tables in the supporting documents should contain footnotes. Recheck all the tables as some columns are not aligned with the entries.

Reviewer 2 Report

Manuscript titled “Novel formula of antiprotozoal mixtures” reports the use of essential oils, organic acids, metal ions and their mixtures as potential antiprotozoal agents. The premise of the work is relevant, since pathogen control remains a challenge in various places around the world, most of which are responsible for a high number of serious illnesses and deaths.

There are some important flaws in the manuscript that should be addressed, among the most significant ones include a lack of proper background data and/or justification for the treatments proposed, an unclear presentation of methods and lack of statistical analysis. Please see detailed comments for specific changes required:

1.       Please add more precise information to the abstract, since it only contains generalities that do not allow to fully determine the findings of the work. For example, organisms tested are not mentioned, nor the most effective oil or mixture against them. Some numerical data to highlight the most important findings may make it more informative. Same comment for the conclusion.

2.       In line 39, the term “symptoms” should be changed to “signs”.

3.       The introduction lacks information to properly justify the current work. For example, lines 94-109 mention scientific interest in studying some essential oils as antiprotozoals, but the following lines (111-113) mention that the work also considered organic acids and metal ions. Thus, please add some information that justifies the authors’ interest in organic acids, metal ions and their mixtures alongside essential oils. Briefly describing some mechanisms of action and their potential synergy against the studied pathogens may be helpful.

4.       Please define the abbreviation “TMMn” in line 137.

5.       Please homogenize the number of decimal digits used in all tables.

6.       Organic acids used are not mentioned in section 4, please include them.

7.       In section 4.4, the proportion of essential oils to acids to metal ions is ambiguously described, since only 5 g of metal salts are mentioned, but not the mass or volume of oils or acids. Also consider specifying the metal ion equivalents added, not just the absolute mass of its salt.

8.       Also in section 4.4, mixtures prepared are not clearly described. Line 119 mentions 48 combinations, but it is not obvious exactly which. A table or diagram would be helpful to illustrate this.

9.       Line 273 mentions “four-fold replicates”, but there is no clear indication of SD or SEM on the results, nor any statistical analysis performed on them.

10.   In line 281, please specify the concentration of chloramphenicol and metronidazole used.

11.   In line 295, a “reaction product” is mentioned. Should this be a mixture instead, or did a chemical reaction actually take place?

Round 2

Reviewer 2 Report

Manuscript titled “Novel formula of antiprotozoal mixtures” reports the use of essential oils, organic acids, metal ions and their mixtures as potential antiprotozoal agents. The work reported is aimed at using a synergistic approach against pathogenic protozoans.

The most recent version of the document was revised according to comments and suggestions made during an initial revision. Among those made by the present reviewer include:

1. Adding more precise information to the abstract and the conclusion, in order to make the findings reported easier to identify and understand. The abstract has been edited to include all agents tested (including organic acids and metal ions), and the most effective combination is now specified. This was also added to the conclusion.

2. Replacing “symptoms” with “signs” (line 43). The change was made.

3. Adding more information to the introduction, which properly justifies and supports the current work. The introduction now contains additional data to justify using the proposed treatments and their combinations.

4. Defining an abbreviation of a treatment used. The abbreviation has been properly defined.

5. Homogenizing the number of decimal digits used in all tables. The number of digits remains to be correctly homogenized to the same precision, for example, in Table 2, chloramphenicol LD50 for E. gracilis is stated as 0.05%, while its LD100 is stated as 0.1%. These values should be provided with the same level of precision, including throughout the main text. Lack of errors for the data further complicates interpreting the effectiveness of the treatments (see comment 9).

6. Mentioning the organic acids used in section 4. They have now been added (section 4.3).

7. Unambiguously describing the proportion of essential oils to acids to metal ions in section 4.4. These have now been described in better detail.

8. Also in section 4.4, providing a clearer description about the mixtures prepared. Authors added a new table (Table 1), but the table is not properly referenced in the main text.

9. Providing a clear indication of SD or SEM on the results, and any statistical analysis that was performed. This has not been properly addressed. It is understandable that adding errors and indications of statistical significance may be excessive for a single table, however, this should be specified somehow, perhaps by splitting the data on additional tables, or as the authors deem more convenient and informative. Furthermore, there is still no indication in the main text that any statistical analysis was actually performed. This is required to objectively state if the treatments are indeed effective or not.

10. Specifying the concentration of chloramphenicol and metronidazole used. Concentration of the stock solutions has now been specified (5 mg/mL), and dilutions are said to have been made until LD values were reached.

11. A “reaction product” is mentioned in line 352. Authors confirm that this is in fact a mixture, but the change was not made on the actual document.

In addition to the aforementioned issues, some are also evident on the new version, such as homogenizing the use of a period or comma to indicate decimal digits (see lines 380 and 381 where both are use contiguously), some minor writing mistakes (see line 20 “The analyzes show…”). Thus, please carefully revise the document to amend these and any other mistakes.
